# High-Performance Power Converter for Charging Electric Vehicles

**Nikolay Madzharov** [1] **and Nikolay Hinov** [2,*]

1 Department of Electronics, Faculty of Electrical Engineering and Electronics, Technical University of Gabrovo, 4 H. Dimitar, 5300 Gabrovo, Bulgaria; madjarov@tugab.bg
2 Department of Power Electronics, Faculty Electronic Engineering and Technology, Technical University of Sofia, 8, Kliment Ohridski Blvd., 1000 Sofia, Bulgaria
\* Correspondence: hinov@tu-sofia.bg; Tel.: +359-2965-2569

**Abstract:** This paper presents the analysis, modeling, simulation and practical studies of resonant inverters with a voltage limitation on the resonant capacitor. The power circuits obtained in this way are characterized by the fact that the power consumption does not depend on the load changes, but is a function of the operating frequency, the value of the resonant capacitor and the supply voltage—these are the so-called inverters with energy dosing. Analytical dependences, simulations and experimental results were determined, which described the behavior of the studied power electronic devices. The obtained expressions for the inverter current in the different stages of the converter operation were the basis for the creation of the engineering methodology for their design and prototyping. Based on the derived basic ratios and characteristics, the capabilities of these devices for self-adaptation to the needs and changes of the load were demonstrated. A comparison of the characteristics of classical resonant inverters and those with energy dosing was made, thus demonstrating their qualities and advantages. The presented results display the properties of this class of circuits and the challenges to their effective application to find the optimal solution for the implementation of charging stations for different specific needs. On the other hand, the limitations in the use of these circuits were that no power was consumed from the power supply during the whole period, the lack of limitation of the maximum current through the transistors and the need for sufficient time to dissipate energy in the resonant inductor when working with high-resistance and low-power loads.

**Keywords:** charging stations; energy dosing; resonant converters; electric vehicles

## 1. Introduction

Mobility is one of the most important characteristics of a modern and smart society. The dynamics and nature of human relations in recent years are unthinkable without the constant movement of people, goods and capital. Environmental problems and challenges have necessitated a new concept for the provision of transport connections and systems through the implementation and growing dominance of electric vehicles. Regarding this aspect, there are several main obstacles to the distribution of electric vehicles: the provision of the necessary electricity, the underdeveloped infrastructure of the electricity transmission network with its insufficient capacity, minimizing the impact of the charging infrastructure on the energy transmission network and the development of a new class of power electronic devices and systems designed to charge energy storage elements [1–3]. Despite the diversity of these research problems, what they have in common is that their sustainable and effective solution is related to the development of power electronics. The aim of the present work was to present the capabilities of a class of power circuits of electronic energy converters, known as resonant converters with energy dosing, which, due to their unique properties and characteristics, are very suitable for the realization of charging stations [4,5].

## 2. Overview of Different Methods and Power Converters for the Realization of Charging Stations

The development of power electronics has led to a wide variety of applications and, accordingly, topologies and operating modes of power electronic converters and systems. In particular, the development of power electronic devices with applications for charging electric vehicles can be divided into two main groups: direct and contactless charging [6–12].

Characteristic of both large groups is the use of resonant transducers in the process of converting electricity. This is due to some of their main advantages over other types of converters, namely [13,14]:

- They work with electrical signals in a shape close to a sinusoid and correspondingly low content of harmonics;
- The realization of soft switching and hence of high efficiency of the devices in a natural way without the participation of additional elements;
- The use of parasitic elements as the main reason for the realization of resonant circuits and processes.

On the other hand, the use of resonant converters is associated with several difficulties and challenges, such as:

- The use of variable frequency control, which creates preconditions for difficult elimination of electromagnetic interference;
- The strong dependence of the operating modes on the tolerances of the building elements that participate in the resonant circuit;
- The incomplete use of power by semiconductor elements.

Addressing these challenges is usually done in two ways: either by improving the methods of and synthesis of power electronic systems control, including the use of artificial intelligence techniques [15–19] or by proposing power schemes that are weakly dependent on the operating modes of the load changes and have possibilities for self-adjustment of the converter to the requirements of the specific application [20–24].

This study proposed the use of resonant converters with improved characteristics that eliminate the most significant part of these shortcomings through the use of voltage-limiting circuits (fully or partially) on the resonant capacitor. These schemes have gained wide popularity in the specialized literature, such as energy dosing schemes. The present research involved the development of the long-term work of the authors in this field and was aimed at the implementation of this class of converters for the purpose of charging stations.

## 3. Basic Relations in the Analysis of Resonant Converters with Energy Dosing

The method of energy dosing (ED) has been used in several applications of AC and DC power supplies, where their use initially began in industrial technologies based on induction heating [4,5,24,25]. The loads of the converters in these applications are characterized by highly variable parameters, which can often vary from idle to short-circuiting. These properties of ED converters make them very suitable for the development of contactless charging stations for static and dynamic charging of electric vehicles. Their stable operation is obtained by fulfilling the following condition:

$$k_S = (dU_T/dI_T - dU/dI) > 0 \tag{1}$$

where $k_S$ is the coefficient of stability of the system, while $dU_T/dI_T$ and $dU/dI$ are the dynamic resistances of the load and the DC-DC converter, respectively.

Figure 1 shows a power scheme of a half-bridge DC-DC converter with energy dosing. It consists of a half-bridge resonant inverter with energy dosing (RI with ED) without reverse diodes, a high-frequency matching transformer and an output rectifier with a capacitive filter and an equivalent load.

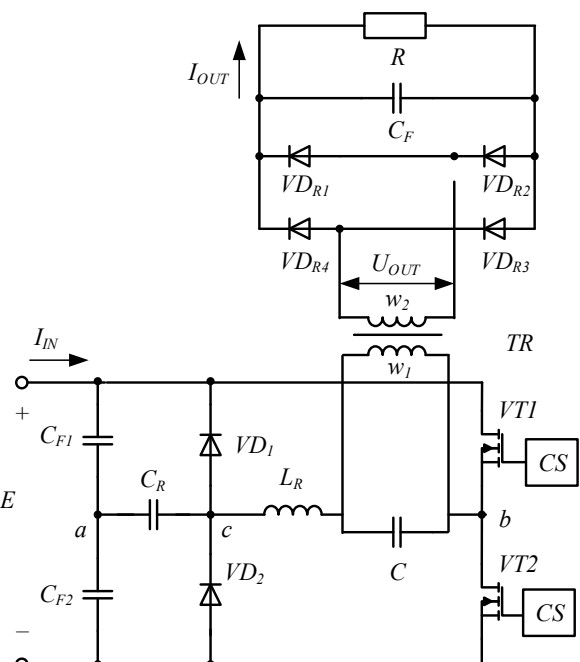

**Figure 1.** Half-bridge DC–DC converter with energy dosing.

Of the possible operating modes, the most suitable when using the circuit is the mode when the operating frequency is less than the resonant frequency of the alternating current (AC) circuit of RI with ED, i.e., $f < f_0$. Figure 2 shows the time diagrams of the current through the resonant inductor $L_R$, the voltage of the resonant capacitor $C_R$, the output current and the voltage of the transistor $VT_1$, which clarifies the principle of operation. It should be noted that the transistors operate with zero on and off current (ZCS).

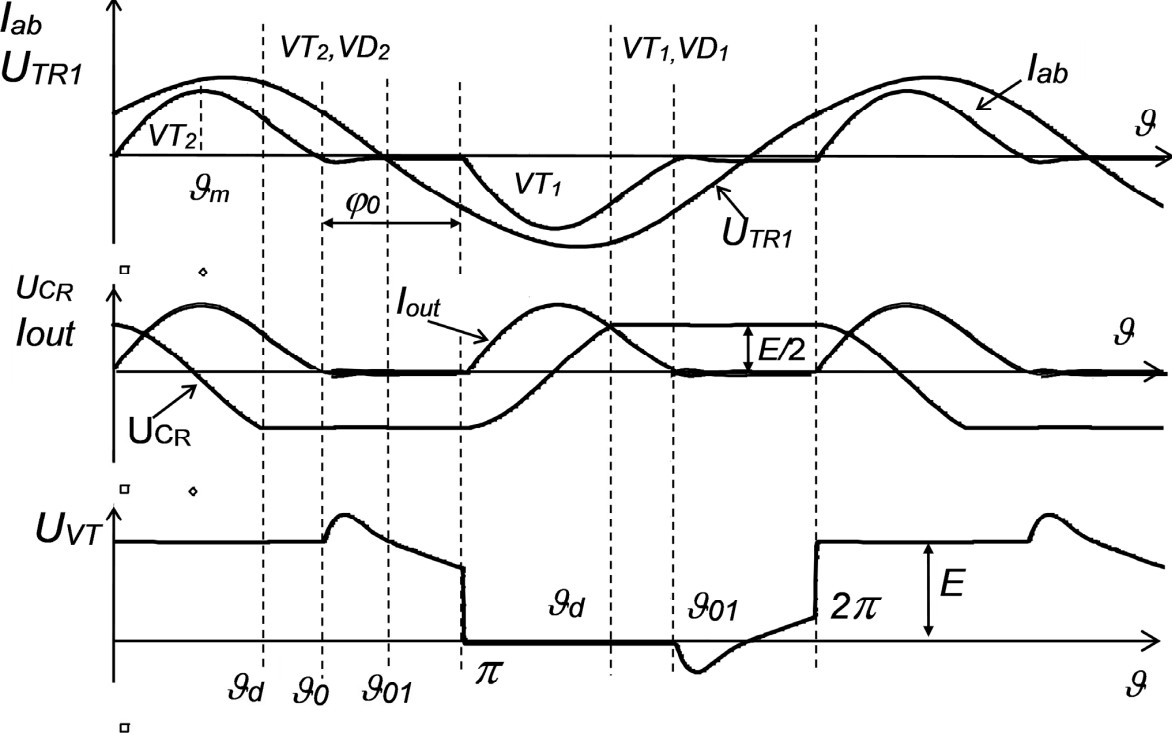

**Figure 2.** Basic timing diagrams of a half-bridge DC–DC converter with energy dosing.

The indicated time diagrams show that in the operation of the power circuit during each half-period the following intervals are distinguished: $0 \div \vartheta_d$, $\vartheta_d \div \vartheta_0$, $\vartheta_0 \div \vartheta_{01}$ and $\vartheta_{01} \div \pi$. On the other hand, for the analysis of the resonant inverter, it is important to distinguish between the intervals $(0 \div \vartheta_d)$ in which energy is consumed from the power source and those in which the energy accumulated in the resonant elements supplies the output of the circuit. The operation of the considered power scheme for one of the two half-periods of its operation $(0 \div \pi)$ is explained in detail using a modal diagram, which is shown in Figure 3. In order to achieve convenient ratios, the following assumptions are made: the active and passive elements are ideal, and the times for their commutation are neglected. In addition, the processes in the scheme are considered after reaching a set mode of operation when there is periodicity and repeatability of the state variables.

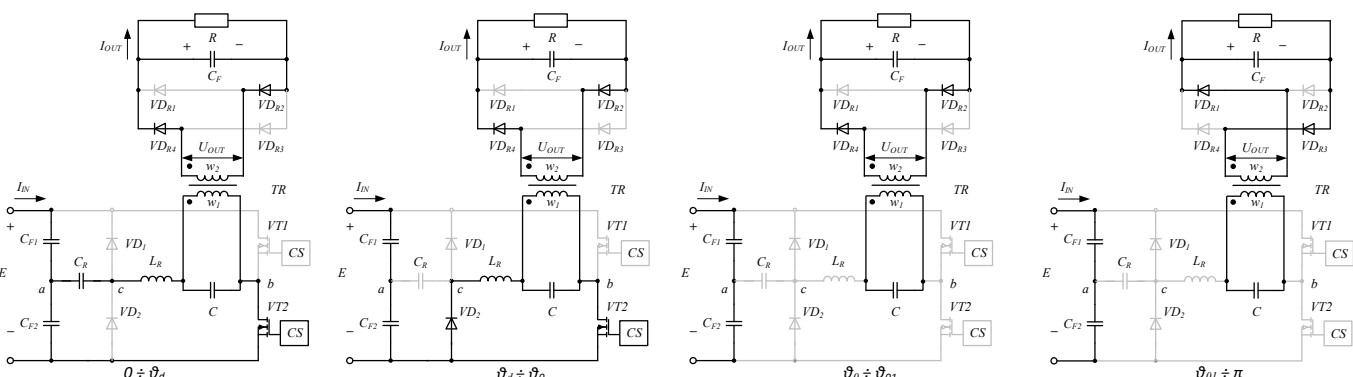

**Figure 3.** Modal diagram of a half-bridge DC–DC converter with energy dosing.

In the analysis of the circuit, the secondary circuit is brought to the primary circuit; as a result of which, the output capacitor, the load and the transformer are replaced by a voltage source $nU_{OUT}$, where $n$ is the transformation coefficient ($n = W_1/W_2$). The behavior of the converter is determined mainly by the values of the elements $L_R$ and $C_R$, and more generally, by the following parameters [5,24,25]:

- Characteristic impedance:

$$Z_0 = \sqrt{L_R/C_R} \tag{2}$$

- Natural resonant frequency of a resonant circuit composed of ideal elements:

$$\omega_0 = \sqrt{1/L_R C_R} \tag{3}$$

- Period of the resonant frequency:

$$T_0 = 2\pi/\omega_0 \tag{4}$$

- Control frequency period—$T$;
- Fill factor (duty cycle):

$$\gamma = T_0/2T \tag{5}$$

If the losses in the elements of the converter are ignored, then its input power will be equal to the output. Taking this assumption into account, the following electrical ratios for the input current and output voltage can be recorded as follows:

$$I_{IN} = I_{OUT}/2n \qquad U_{OUT} = E/2n \tag{6}$$

Diodes $VD_1$ and $VD_2$ start to conduct when the maximum and minimum values of the voltage on the capacitor $C_R$ becomes equal to $+E/2$ or $-E/2$. If the load resistance or operating frequency has values that are significantly different from the nominal ones, the capacitor $C_R$ will not be recharged from $-E/2$ to $+E/2$ and the diodes $VD_1$ and $VD_2$

will not turn on. In this case, the voltage to which the capacitor will be recharged can be determined using the expressions from Equations (2) to (6).

$$U_{CR} = E/4 + \pi E Z_0/8\gamma R n^2 \tag{7}$$

The latter expression has a very important practical meaning. It displays the ratio for the values of the load resistance at which the energy dosing mode is performed, i.e., limiting the voltage on the capacitor $C_R$ to $\pm E/2$ by $VD_1$ and $VD_2$, namely,

$$R \leq \pi Z_0/2\gamma n^2 \tag{8}$$

This mode, which is considered optimal, is characterized by two main intervals: the consumption of energy from the power supply and the short circuit of the alternating current circuit when energy from the power supply is not consumed.

*3.1. Electromagnetic Analysis of the Converter at the Time of Energy Consumption from the Power Source $0 \div \vartheta_d$*

During the interval $0 \div \vartheta_d$ (Figure 2), only the transistor $VT_2$ is unlocked and the equivalent circuit consists of the power supply connected in series, the reduced load circuit and the resonant inductor and capacitor. The natural frequency of the AC circuit (a-b in Figure 2) must be higher than the control frequency, i.e., $\omega_0/\omega > 1$, in order to obtain a tendency for the current to drop to zero at the beginning of the interval $\varphi_0$. The following frequency ratio is valid for the considered operating mode:

$$\frac{\omega_0}{\omega} = \frac{\pi}{\pi - \varphi_0 - (0.65 \div 1.7)\varphi_0} > 1 \tag{9}$$

$$\text{i.e., } \omega_0/\omega = 1.2 \div 1.4 \tag{10}$$

Taking into account the expression for the current through the resonant inductor:

$$i_{L_R}(\vartheta) = \frac{E}{\omega_0 L_R} \exp^{-\vartheta/2Q} \sin \frac{\omega_0}{\omega} \vartheta \tag{11}$$

and the voltage across the resonant capacitor $C_R$:

$$u_{CR}(\vartheta_d) = \frac{1}{\omega_0 C_R} \int_0^{\vartheta_d} i(\vartheta) = \frac{E}{2} \tag{12}$$

as determined at the end of the interval of energy consumption from the power source, the moment that corresponds to the angle $\vartheta_d$ is

$$\vartheta_d = [\pi - arctg(2Q\omega_0/\omega)]/(\omega_0/\omega) \tag{13}$$

where $Q = \omega L_R/R_E$ is the quality factor of the resonant circuit, $R_E$ is the equivalent resistance of the AC circuit between points b and c of Figure 1 and $\vartheta = \omega t$.

*3.2. Operation of the Converter in the Short Circuit Interval of the Alternating Current Circuit $\vartheta_d \div \vartheta_0$*

At the moment corresponding to the angle $\vartheta_d$, the resonant capacitor $C_R$ is charged to voltage $-E/2$ and the diode $VD_2$ is turned on. In essence, the electromagnetic processes in the second interval $\vartheta_d \div \vartheta_0$ are aperiodic. The expression for the current through the resonant inductor is of the form

$$i_{LR}(\vartheta) = i_{LR}(\vartheta_d) \exp^{-\frac{R}{\omega L_R}\vartheta} \tag{14}$$

where $R = R_E$.

It is very important from a practical point of view to determine the value of the current at the end of the interval (moment corresponding to an angle $\pi - \varphi_0$) because the level of switching losses in the transistors depends on it. It is calculated using the current expression $i_{LR}(\vartheta)$ at $\vartheta = \pi - \varphi_0$. In this regard, additional information about the processes in the circuit is given by the expressions for the voltages across the resonant inductor and the equivalent load:

$$u_{LR} = -L di_{LR}/dt \tag{15}$$

$$u_{OUT} = R\left[n i_{LR}(\vartheta_d) \exp^{-\vartheta/Q}\right] \tag{16}$$

From the presented relations, it is clear that if at the moment corresponding to an angle $\pi - \varphi_0$, the resonant current has zero value, then the voltages $U_{LR}$ and $U_{OUT}$ will also have zero values. At this point, the voltage on the transistors will also have a zero value, i.e., they will turn on at zero current and will turn off at zero current and zero voltage.

### 3.3. Stabilization and Regulation of the Output Power and Voltage of the Converter

The energy of the capacitor when recharging from $-E/2$ to $+E/2$ is equal to

$$W = C_R E^2/2 \tag{17}$$

When transistors $VT_1$ or $VT_2$ are turned on, this energy is transmitted to the load, i.e.,

$$C_R E^2/2 = U_{OUT}\, I_{OUT}\, T/2. \tag{18}$$

In this case, the power of the converter $P$ for one period is expressed by the ratio

$$P = E^2 f C_R = E\, I_d = U_{OUT}\, I_{OUT} = const, \tag{19}$$

The first conclusion that can be drawn from the expressions for $W$ and $P$ is that at a constant operating frequency, supply voltage and capacitance of the resonant capacitor, the transmitted power in the load is constant and does not depend on its parameters. Maintaining a constant power means that the output DC voltage $U_{OUT}$ is self-aligning with the load parameters.

To set the power level in accordance with Equation (19), the value of the capacitor $C_R$ is most often changed. For this purpose, the developed electronic keys are used, for which there are author's claims regarding their use for similar purposes. They consist of only one transistor (IGBT or MOSFET) with a reverse diode and a series-connected capacitor $C_R$, the capacitance of which is in accordance with the desired power (Equation (19)). Figure 4 shows the schematic diagram of four electronic switches, which participate in the circuit of the converter of Figure 1 and are connected to points a and c. The presented circuit contains half as many elements as the traditional electronic switch, while filled with two oppositely connected transistors with reverse diodes and, therefore, has less static losses compared to other embodiments [24,25].

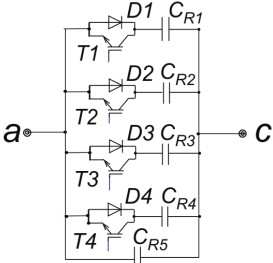

**Figure 4.** Electronic semiconductor switches for setting the power level.

The combinations between the turned on and turned off capacitors make it possible to set fifteen power values in the range $0 \div P_{NOM}$. No galvanically separated signals are required to control the individual transistors.

It is important to note that the switching of the capacitors $C_{R1}$–$C_{R4}$ from Figure 4 can be done during the operation of the converter without receiving current and voltage overloads from the transistors.

The second characteristic feature of the converter is obtained by substituting the expression for the load current $I_{OUT} = U_{OUT}/R$ into Equation (19). In this way, a ratio is obtained that shows the relationship between the input and output voltage:

$$U_{OUT} = E\sqrt{C_R.R/T} \tag{20}$$

The conclusion to be drawn from this expression is that by changing the operating frequency, the output voltage can be invariably maintained when the value of the load and/or the input voltage changes. Figure 5 shows the dependence of the output voltage in relative units as a function of frequency at different loads. The information from this characteristic is used in the design of the converter because it takes into account, on the one hand, the relationship between the value of the load and the capacitance of the capacitor $C_R$ setting the power and, on the other, the dependence of output voltage on frequency and input voltage.

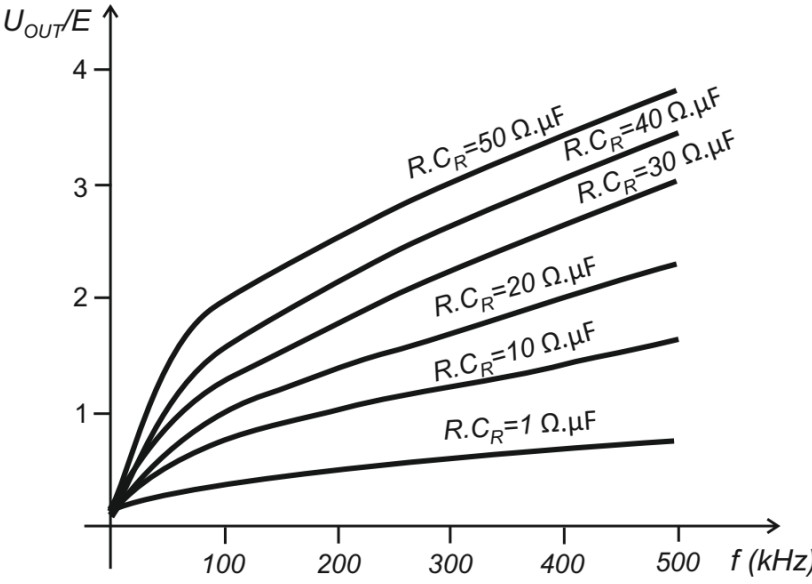

**Figure 5.** Control characteristics of a converter with energy dosing.

The regulation or stabilization of the output voltage is carried out via feedback, changing the operating frequency of the converter. The analytical dependence of the regulation law can be deduced by differentiating the output power expression with respect to the control frequency $f$. After some transformations using the expressions:

$$U_{OUT}\, I_{OUT} = C_R E^2 f, \tag{21}$$

$$U_{OUT}\, di_{OUT} + I_{OUT}\, du_{OUT} = C_R E^2 df, \tag{22}$$

$$du_{OUT} = di_{OUT}(R//1/\omega\, C_F) \tag{23}$$

$$I_{OUT} = U_{OUT}/R \tag{24}$$

it is found that $\dfrac{du_{OUT}}{df} = \dfrac{E}{2}\sqrt{\dfrac{C_R R}{f}}\,\dfrac{1}{1+\omega C_F R/2}.$ $\tag{25}$

This expression represents the transfer function of the control system and represents the law according to which the operating frequency must be changed when the load parameters change in order to stabilize the constant output voltage.

## 4. Investigation and Comparison of the Characteristics and Converters with and without Energy Dosing Used in the Fast Charging Stations of Electric Vehicles

Based on the analytical expressions obtained in the analysis of the studied scheme (Figure 1), its characteristics were obtained. For greater objectivity of the formulated conclusions, the characteristics of RI with ED without reverse diodes (RD) of power transistors (PT) and two other competing schemes that are often used in charging stations as high-frequency sources [20–22,26–29]: a half-bridge RI with RD and a full-bridge parallel current fed inverter (PCFI). The use of the same input data for the indicated schemes allows for objective comparisons and definitions of their advantages and disadvantages. In Tables 1–3 and the numerical values from the research are presented, as in the column "Type", the conditions under which the results are obtained are given. In these studies, emphasis was placed on the behavior of the circuits when changing the equivalent load parameters, which was achieved by varying the air gap of the contactless transmitter and/or the load resistance. Specifically for each scheme, they were the following:

- For the RI with ED—"0"—nominal mode, the following tests were done when the load parameters change compared to the nominal ones under the following conditions: "1"—without the RD of the PT and the operating frequency equal to the nominal one; "2"—with the RD of the PT and the operating frequency equal to the nominal; "3"—with the phase between the current and the voltage of the parallel load circuit, other than for a pause $\varphi_0$ when changing the operating frequency; and "4"—with the phase between the current and the voltage of the parallel load circuit equal to the pause $\varphi_0$ when changing the operating frequency.
- For the RI with RD—"0"—nominal mode, the following tests were done when the load parameters changed compared with the nominal ones under the following conditions: "1"—at the operating frequency equal to the nominal one and "2" and "3"—with a change in operating frequency;
- For the PCFI—"0"—nominal mode, the following tests were done when the load parameters changed compared to the nominal ones under the following conditions: "1"—at the operating frequency equal to the nominal one and "2"—with a change in the operating frequency.

The main conclusions that can be drawn were about the reliable operation of the RI with ED, which for these modes, was determined by the conditions for the switching PT. The most favorable were those in which the current and/or voltage were zero (ZCS, ZVS, ZCSZVS). Depending on the equivalent parameters of the AC circuit, the converters operated in one of three modes. If the switching mode at zero current was conditionally accepted as nominal (basic) due to the change of the load or the operating frequency, different changes in the values of the main electrical and phase ratios were observed.

**Table 1.** Characteristics of the RI with ED.

| Resonant Inverter with Energy Dosing, Characteristics at the Following Element Values: $C_{R1} = C_{R2} = 1.5\ \mu F,\ L_R = 11.1\ \mu H,\ C = 26\ \mu F$ | | | | | | | | | | | | | | | | | | | |
|---|---|---|---|---|---|---|---|---|---|---|---|---|---|---|---|---|---|---|---|
| $R_T$ (Ω) | | 0.025 | | | 0.0375 | | | 0.0425 | | | 0.0575 | | | ***0.05*** | | 0.0625 | | | 0.075 | |
| $L_T$ (μH) | | 1.15 | | | 1.725 | | | 1.955 | | | 2.645 | | | ***2.3*** | | 2.875 | | | 3.45 | |
| Type | 1 | 2 | 3 | 4 | 1 | 2 | 3 | 4 | 1 | 2 | 3 | 4 | ***0*** | 1 | 2 | 3 | 4 | 1 | 2 |
| $P$ (kW) | 15 | 4 | 15 | 20 | 15 | 8.5 | 15 | 18 | 15 | 12 | 15 | 17 | ***15*** | 15 | 9.9 | 14.6 | 14.5 | 15 | 10.6 |
| $f$ (kHz) | 20 | 20 | 27 | 28.3 | 20 | 20 | 22 | 23.5 | 20 | 20 | 21.2 | 22 | ***20*** | 20 | 20 | 19.5 | 19.6 | 20 | 20 |
| $U_{Cm}$ (V) | 73 | 73 | 224 | 270 | 152 | 149 | 216 | 256 | 192 | 191 | 231 | 248 | ***236*** | 252 | 209 | 224 | 225 | 258 | 217 |
| $I_M$ (A) | 280 | 280 | 151 | 148 | 182 | 183 | 154 | 165 | 161 | 162 | 152 | 162 | ***159*** | 213 | 175 | 159 | 158 | 251 | 210 |
| $I_{ON}$ (A) | 0 | 0 | 0 | 0 | 0 | 0 | 0 | 0 | 0 | 0 | 0 | 0 | ***0*** | 0 | 0 | 0 | 0 | 0 | 0 |
| $I_{OFF}$ (A) | 207 | 209 | 120 | 50 | 140 | 146 | 79 | 0 | 97 | 90 | 41 | 0 | | 0 | 0 | 8 | 0 | 0 | 0 |

**Table 2.** Characteristics of the RI with an RD.

| Resonant Inverter with Reverse Diodes, Characteristics at the Following Element Values: $L_R$ = 28 µH, $C_{R1}$ = $C_{R2}$ = 0.86 µF, C = 15 µF, $C_F$ = 2000 µF | | | | | | | | | | | | | | | | | | | | |
|---|---|---|---|---|---|---|---|---|---|---|---|---|---|---|---|---|---|---|---|
| $R_T$ (Ω) | 0.045 | | | | 0.675 | | | 0.0765 | | | ***0.09*** | | 0.1035 | | | 0.1125 | | | 0.135 | |
| $L_T$ (µH) | 2.05 | | | | 3.075 | | | 3.485 | | | ***4.1*** | | 4.715 | | | 5.125 | | | 6.15 | |
| Type | 1 | 2 | 3 | 0 | 2 | 3 | 1 | 2 | 3 | ***0*** | 1 | 2 | 3 | 1 | 2 | 3 | 1 | 2 | 3 |
| P (kW) | 20.4 | 15 | 15 | 15 | 15 | 16.4 | 33.1 | 15 | 15 | ***15*** | 8.4 | 15 | 15 | 6.5 | 15 | 15 | 3.6 | 10.5 | 15 |
| f (kHz) | 20 | 19.8 | 23.7 | 20 | 18.2 | 23.5 | 20 | 22.3 | 17.6 | ***20*** | 20 | 18.2 | 23.5 | 20 | 17 | 23.5 | 20 | 15.5 | 23.9 |
| $U_{Cm}$ (V) | 2510 | 2251 | 1293 | 677 | 1620 | 656 | 1252 | 656 | 1486 | ***919*** | 670 | 739 | 1547 | 738 | 827 | 1711 | 804 | 800 | 1800 |
| $U_{ОБP}$ (V) | 249 | 209 | 249 | 305 | 235 | 321 | 411 | 306 | 246 | ***548*** | 242 | 289 | 400 | 222 | 288 | 395 | 178 | 241 | 355 |
| $I_M$ (A) | 482 | 420 | 272 | 136 | 270 | 107 | 207 | 109 | 238 | ***31*** | 101 | 111 | 327 | 104 | 126 | 370 | 127 | 109 | 393 |
| $I_{ON}$ (A) | 450 | 390 | 0 | 15 | 226 | 0 | 0 | 18 | 187 | ***28*** | 54 | 14 | 311 | 69 | 30 | 350 | 109 | 33 | 370 |
| $I_{OFF}$ (A) | 0 | 0 | 270 | 0 | 0 | 0 | 45 | 0 | 0 | ***28*** | 0 | 0 | 0 | 0 | 0 | 0 | 0 | 0 | 0 |

**Table 3.** Characteristics of the PCFI.

| Full-Bridge Parallel Current Fed Inverter, Characteristics at the Following Element Values: $L_1$ = 1.6 mH, C = 1.73 µF | | | | | | | | | | | | | |
|---|---|---|---|---|---|---|---|---|---|---|---|---|---|
| $R_T$ (Ω) | 0.432 | | 0.648 | | 0.743 | | ***0.86*** | | 0.994 | | 1.08 | | 1.296 | |
| $L_T$ (µH) | 19.9 | | 29.85 | | 33.83 | | ***39.8*** | | 45.87 | | 49.75 | | 59.7 | |
| Type | 1 | 2 | 1 | 2 | 1 | 2 | ***0*** | 1 | 2 | 1 | 2 | 1 | 2 |
| P (kW) | 127.5 | 15 | 25.5 | 15 | 13 | 15 | ***15*** | 32.5 | 15 | 50.5 | 15 | 99 | 15 |
| f (kHz) | 20 | 26.05 | 20 | 23 | 20 | 21.65 | ***20*** | 20 | 18.7 | 20 | 18 | 20 | 16.5 |
| $U_{Cm}$ (V) | 1800 | 876 | 1100 | 922 | 816 | 923 | ***919*** | 1455 | 924 | 1919 | 935 | 2894 | 938 |
| $U_{ОБP}$ (V) | - | - | - | - | 0 | 500 | ***548*** | 1248 | 559 | 1800 | 550 | 2875 | 551 |
| $I_M$ (A) | 255 | 31 | 52 | 31.2 | 27 | 31.2 | ***31*** | 68 | 31 | 103 | 31.7 | 198 | 31.7 |
| $I_{ON}$ (A) | 250 | 29.5 | 50 | 28.9 | 26 | 29 | ***28*** | 62 | 28 | 95 | 27.5 | 186 | 28 |
| $I_{OFF}$ (A) | 250 | 29.5 | 50 | 28.9 | 26 | 29 | ***28*** | 62 | 28 | 95 | 27.5 | 186 | 28 |

The other conclusions were as follows:

(1) As the load parameters (R) decreased, the load circuit and the equivalent AC circuit gained an inductive misalignment. For this reason, the frequency of the RI's own oscillations decreased. The current of the PT, when it was switched off by the control pulses, was different from zero, and this created conditions for switching on the reverse diode of the inoperative transistor. The described phenomenon is the reason for the partial reduction of the RI power, as the second recharging of the resonant capacitors in the considered half-period began. As the operating frequency increased, the power could be set equal to the nominal one, and at the same time, the effect of reducing the switching current of the PT-$I_{OFF}$ was obtained. When the resonance in the load oscillating circuit was reached again, both currents $I_{OFF}$ and $I_{ON}$ became equal to zero. Such a favorable development of the processes was observed when reducing the load parameters to 25–30%. With an even greater reduction in the load, for example, by 50% to achieve $I_{ON} = I_{OFF} = 0$, it was necessary to increase the frequency, which led to an increase in power by 30% above the nominal value (Table 1).

(2) When the inverter was unloaded (R increased), the load oscillating circuit was capacitively disrupted. The frequency of the inverter's natural oscillations increased and the current of the PT naturally became equal to zero before the termination of the control pulse. For this reason, conditions were created for the occurrence of a mode with a natural shutdown of the PT until the power drops, as noted above. By reducing the operating frequency until the resonance in the load oscillating circuit was restored, the ZCS mode could be re-established ($I_{ON} = I_{OFF} = 0$), but the power, although insignificant, was reduced (by 5–10%) in accordance with Equation (19).

From the point of view of the operational characteristics and reliability of RI with ED, it is important that in all operating modes, it is not possible to exceed the set power determined by Equation (19). In practice, energy dosing protects the PT from overload and ensures the natural adaptation and self-alignment of the RI to the load. This, in turn, determines a major positive quality of an RI with ED that is not possessed by a traditional RI.

In the case of similar changes in the load, the circuit of the RI with ED without an RD of the PT always worked with the nominal power without the need to change the operating frequency. The technical applicability of this circuit variant was determined mainly by the voltage on the PT, which was obviously greater than that in the RI with ED and the OD of the PT.

Another important result of the presented characteristics should be noted, which was that switching the PT currents on and off in both cases (when increasing and decreasing R) could be maintained equal to zero if the operating frequency changes in one direction or another until resonance was obtained in the load circle (Table 1). The fact that the most favorable modes in terms of power and currents of switching on and switching off of the PT were obtained by the resonance of the load circuit, facilitates the synthesis of control of the power electronic converter.

The second group of conclusions was about the qualities of the other two schemes and their comparisons with the RI with ED: (1) In the half-bridge RI with RD, with the change of the load parameters, the power changed more strongly (Table 2), and to return to the nominal value, it was necessary to regulate the frequency more deeply, which significantly increased the voltage on the resonant capacitors $U_{CRm}$ and currents on and off (i.e., the PT-$I_{ON}$ and $I_{OFF}$). Generally speaking, this scheme could not cover a range of variation of the load parameters greater than 15–20% while maintaining a favorable operating mode of the PT. (2) In the full-bridge PCFI, there was an even stronger change in power and the need for deeper regulation of the frequency for its stabilization (Table 3). As R increased, the voltage of the oscillating circuit $U_{Cm}$ increased and, accordingly, the forward and reverse voltages of the semiconductor devices increased. In some of the modes, the circuit time for PT recovery disappeared. Based on the presented results, it was concluded that both alternative RIs had significantly less coordination (regulatory) capabilities than the RI with ED.

A visual comparison is shown in Figure 6, where the characteristics of the output power changing for the load parameter variations for the full-bridge-current-fed inverter (curve 1), half-bridge RI (curve 2), RI with ED and reverse diodes (curve 3) and RI with ED without reverse diodes (curve 4).

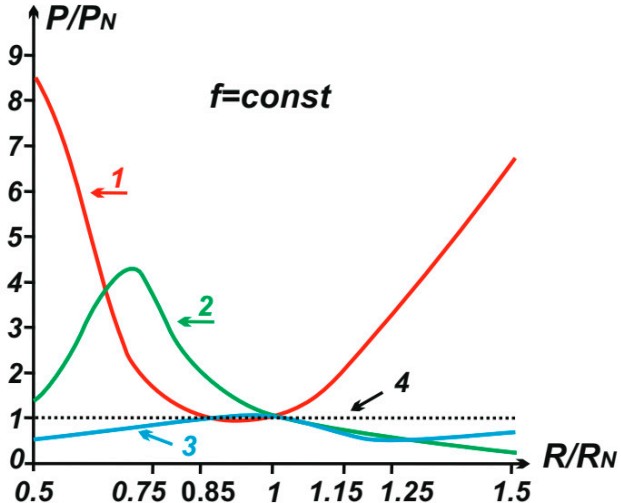

**Figure 6.** Comparison of the output characteristics P/PN = f(R/RN) of the full-bridge-current-fed inverter (1), half-bridge RI (2), RI with ED and reverse diodes (3) and RI with ED without reverse diodes (4).

An important point here is that the last two characteristics (curves 3 and 4 in Figure 6) guaranteed practical adaptively of the inverter to the load and its changes, due to which, it could also operate without regulation when the load changes within the limits stated above, i.e., from 0.5 to 1.5 times relative to the nominal value. This contributes to a significant

improvement in the technical and operational qualities of the high-frequency source and provides the opportunity to work with a wide range of loads, which is typical for contactless charging stations for motorcycles and electric vehicles. Based on these characteristics, it was concluded that energy dosing schemes are advantageous over other types of resonant converters.

On the other hand, according to the characteristics of Figure 6 for the most frequently used topologies based on resonant inverters with reverse diodes (most often LLC), the conclusions made in [30–32] regarding the need to use optimal control in order to realize the charging cycles were confirmed.

## 5. Experimental Results

With the methodology thus created, a full-bridge RI with ED, used for a contactless charging station of a motoped, was designed and studied. It is shown in Figure 7 and consisted of the following main blocks: an electronic converter (inverter), together with the control system and drivers; a power supply; a measuring system; and a load composed of a transmitter and receiver. The power circuit of the inverter was implemented according to Figure 1.

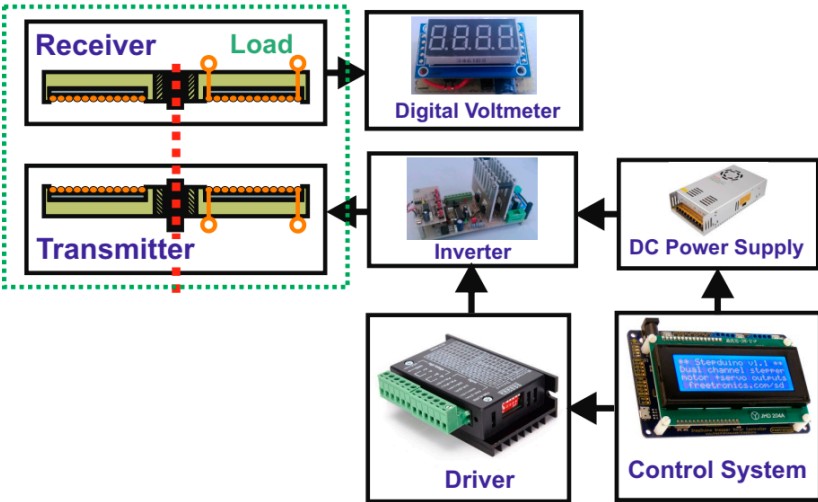

**Figure 7.** Block diagram of the experimental setup.

During the implementation of the experiment, the following input data were used: output power P = 2.5 kW, operating frequency of the inverter f = 30 kHz, power supply voltage E = 300 V, resonant capacitor $C_R$ = 0.95 μF, resonant inductor $L_R$ = 16.31 μH, $W_1/W_2$ = 1, $\omega_0/\omega$ = 1.2–1.4 and equivalent load R = 8.9 Ω.

Figure 8 shows a photo of the stand on which the experimental results were obtained.

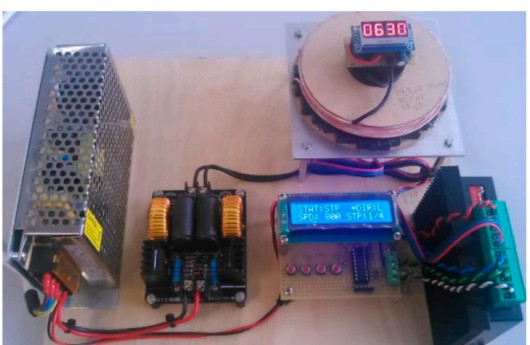

**Figure 8.** A PI with ED that was used for the realization of a contactless charging station of a motoped.

Furthermore, simulation studies of the system were performed with these data. In Table 4, the results obtained from the stand are compared with those from a program for the simulation of power electronic devices. The table shows that RI actually had dosing properties, which was determined by the average value of the consumed current $I_0$, equal to the input current $I_{in} = P/E = 8.3$ A, which was practically the same value from the analysis and the computer experiment.

**Table 4.** Results from the design, the computer experiment and the practical research.

| Value | $\vartheta_d$, (°) | $I_{in}$ (A) | $U_{out}$ (V) | $I_{out}$ (A) | $I_{VTm}$ (A) | $I_{VT}$ (A) | $I_{VD}$ (A) |
|---|---|---|---|---|---|---|---|
| Calculated | 92.1 | 8.3 | 148 | 16.6 | 42.9 | 9.3 | 1.02 |
| Simulation | 93.5 | 8.1 | 145 | 16.2 | 41.1 | 9.1 | 1 |
| Stand | 95 | 8.5 | 147 | 16.5 | 43.6 | 9.6 | 1.1 |

This basic property of the circuit was also confirmed by the load voltage $U_{outm}$, which also coincided with the calculations and the experiment. Obtaining the set power could be shown by the following result of the table, corresponding to the physical processes in RI with ED, namely, that the difference between the currents of the transistors and diodes, i.e., $I_{VT}$ and $I_{VD}$, were equal to $I_{in}$ and corresponded to the energy consumption of energy in the interval $\vartheta = 0 \div \vartheta_d$ and a short circuit of the supply DC bus in the interval $\vartheta = \vartheta_d \div \pi - \varphi_0$. In addition, compliance with the boundary conditions for switching and periodicity could be used as an additional criterion for the reliability of the obtained results. In this case, they were expressed as obtaining the set operating mode of the RI in which the PT turned on and off at zero current (ZCS) (see Figure 9a). It should also be noted that in the calculations, the parameters of the current pulse through the key devices were obtained with great reliability, as the difference with the measured values did not exceed 5%.

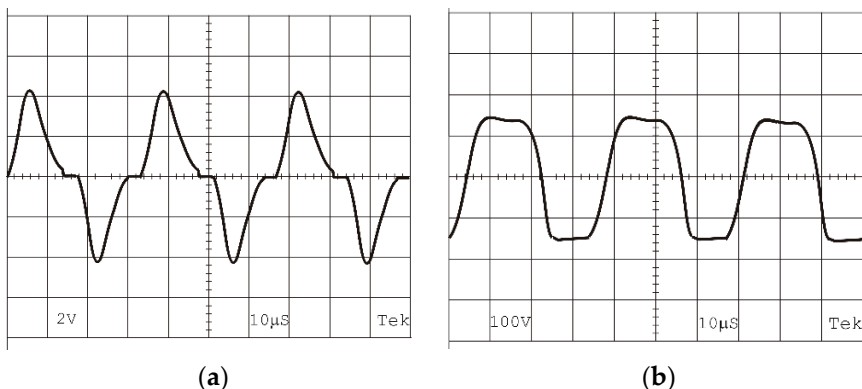

(**a**)                         (**b**)

**Figure 9.** Graphical results obtained from the operation of the stand: (**a**) current through the resonant inductor (43.6 A maximum value) and (**b**) voltage on the resonant capacitor (maximum value 150 V).

Figure 9 presents the results of the following measurements: (a) current through the resonant inductor and (b) voltage on the resonant capacitor, performed with the help of the experimental bench under the conditions described above and operating frequency 30 kHz.

The presented experimental results fully supported the conclusions made in the analysis of the power scheme and confirmed the validity of the design methodology.

## 6. Discussion

The main results obtained from the theoretical and applied scientific problems developed in this article were as follows:

(1) A methodology for designing an RI with ED was developed that provided switching of semiconductor devices at ZCS and ZVS and was characterized by a satisfactory accuracy of not less than 5%, as shown by computer and practical experiments.

(2) A study of the current pulse of the RI was made and its parameters were determined in order to obtain the minimum installed reactive power in the alternating current circuit and electrical loads and losses in the transistors.

(3) The property of the RI with ED was shown to maintain a constant power in the load, regardless of the change of its parameters, when switching the semiconductor devices at ZCS and ZVS. This adaptability and self-matching properties make it very flexible and convenient to use as a wide-range power source for charging stations, including contactless charging.

From the comparisons made with studies of power circuits used for the realization of charging stations presented in [27,28,30–32], it is necessary to conclude that the main alternative of the considered converters with energy dosing was the resonant inverters with reverse diodes and voltage-fed inverters. In order to work with soft commutations, the latter requires the addition of additional resonant circuits (so-called quasi-resonant), and in the case of reverse diode circuits, it is possible to work both in soft commutation mode and to limit the maximum current of the transistors [33]. Unfortunately, in inverters with reverse diodes, power maintenance is achieved through the synthesis of complex control algorithms and controllers. On the other hand, the challenges and limitations associated with the implementation of an RI with ED are related to the fact that no energy is consumed from the power source during the whole period, the lack of limitation of the maximum current through the transistors and the need for sufficient time to dissipate the energy in the resonant inductor when working with high-resistance loads and low power. Regarding this aspect, the achievement of electromagnetic compatibility standards requires the addition of additional modules and ancillary devices.

## 7. Conclusions

The manuscript presents a study of a charging station for electric vehicles based on a resonant converter with energy dosing. Based on the analysis performed in the established mode of operation of the power circuit, the main relations were derived, through which, the values of the circuit elements were determined. On the basis of the analytical expressions, computer simulations and experiments, the advantages of this class of schemes for the realization of charging stations with different capacities and applications were shown. Regarding a continuation of the current research, the combination of a design with techniques of mathematical modeling and computational mathematics in order to determine optimal values of circuit elements for different objective functions, such as minimum losses, maximum efficiency and minimum dimensions, should be undertaken. In addition, future research could conduct experiments at higher capacities, and in order to achieve greater flexibility, they will be built on a modular principle, as well as the dynamic performance of the system, in order to achieve an aperiodic transition process with minimum duration.

The proposed scheme was also successfully used for contactless dynamic charging of electric vehicles [34]. The main advantage of energy dosing schemes is that the power does not depend on the size of the load. Therefore, during driving, when the equivalent load is constantly changing (due to the coefficient of magnetic coupling), the power transferred to the vehicle is constant.

**Author Contributions:** N.M. and N.H. were involved in the full process of producing this paper, including conceptualization, methodology, modeling, validation, visualization and preparing the manuscript. All authors have read and agreed to the published version of the manuscript.

**Funding:** This research was funded by Bulgarian National Scientific Fund, grant number КП-06-H37/25/18.12.2019, and the APC was funded by КП-06-H37/25/18.12.2019.

**Institutional Review Board Statement:** Not applicable.

**Informed Consent Statement:** Not applicable.

**Data Availability Statement:** Not applicable.

**Acknowledgments:** This research was carried out within the framework of the project "Optimal design and management of electrical energy storage systems", KΠ-06-H37/25/18.12.2019, Bulgar-ian National Scientific Fund.

**Conflicts of Interest:** The authors declare no conflict of interest.

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
