# Peer review of "High-Performance Power Converter for Charging Electric Vehicles"

_energies, doi:10.3390/en14248569_

Round 1
Reviewer 1 Report
Have a nice day
I read your manuscript, it was good in present, but needs some improving. such as abstract structure, should be as following:-
The significance of work.
Problem statement.
Objective.
Methodology.
Results.
Outcome.
All above in short sentences.
Regards
Author Response
First of all, we would like to thank you for the thorough review of our paper (energies-1465722) and the useful remarks to improve it.
Reviewer 1
Comments to the Authors
I read your manuscript, it was good in present, but needs some improving. Such as abstract structure, should be as following:-
The significance of work.
Problem statement.
Objective.
Methodology.
Results.
Outcome.
All above in short sentences.
To Reviewer 1:
Thank you very much for your review and valuable remarks.
Such as abstract structure, should be as following:-
The significance of work.
Problem statement.
Objective.
Methodology.
Results.
Outcome.
All above in short sentences.
- Thank you very much for the comments and remarks. The abstract as well as the rest of the text of the manuscript have been revised and edited.
Thank you on behalf of all authors for the accurate and exact review of the our manuscript!
Reviewer 2 Report
1.There are too few new high-quality papers in the references
2.From the experimental results, the proposed method has good results, but the experiment is lack of comparison with other latest schemes in the same field.
3.The modal diagram corresponding to each moment in figure 1 is lacked, which can significantly explore the working principle combined with the text analysis in section
4. 208 and 209 lines of formulas are garbled, and they should be re-typed according to the required format.
5. The full name of some abbreviations should be given. For example, ESR and DCM in the text.
6. The analysis of the results is relatively simple and mostly stays on the surface, without in-depth study of the reasons, including but not limited to: the explanation of Figure 5 simply summarizes that the non-ideal converter has a higher cut-off frequency than the ideal converter ,Did not explain the main reason for this difference.
Author Response
Reviewer 2
Comments to the Authors
1.There are too few new high-quality papers in the references
2.From the experimental results, the proposed method has good results, but the experiment is lack of comparison with other latest schemes in the same field.
3.The modal diagram corresponding to each moment in figure 1 is lacked, which can significantly explore the working principle combined with the text analysis in section
- 208 and 209 lines of formulas are garbled, and they should be re-typed according to the required format.
- The full name of some abbreviations should be given. For example, ESR and DCM in the text.
- The analysis of the results is relatively simple and mostly stays on the surface, without in-depth study of the reasons, including but not limited to: the explanation of Figure 5 simply summarizes that the non-ideal converter has a higher cut-off frequency than the ideal converter ,Did not explain the main reason for this difference.
To Reviewer 2:
Thank you for your review and valuable remarks.
1.There are too few new high-quality papers in the references
- The manuscript has been edited. New reference on the subject from the last year has been added.
2.From the experimental results, the proposed method has good results, but the experiment is lack of comparison with other latest schemes in the same field.
- The main purpose of the manuscript is to present the capabilities of the resonant inverter with energy dosing for self-alignment of the load and for power regulation in a wide range. In the literature, including the added new one, there are enough comparisons of different topologies (mostly resonant). In addition, the characteristics of the basic schemes used for the realization of charging stations are compared in Fig. 6, where additional explanations are added.
3.The modal diagram corresponding to each moment in figure 1 is lacked, which can significantly explore the working principle combined with the text analysis in section
- Thank you very much for the recommendation! A new figure 3 has been added.
- 208 and 209 lines of formulas are garbled, and they should be re-typed according to the required format.
- The necessary adjustments have been made.
- The full name of some abbreviations should be given. For example, ESR and DCM in the text.
- The necessary adjustments have been made.
- The analysis of the results is relatively simple and mostly stays on the surface, without in-depth study of the reasons, including but not limited to: the explanation of Figure 5 simply summarizes that the non-ideal converter has a higher cut-off frequency than the ideal converter ,Did not explain the main reason for this difference.
- The manuscript has been edited. An explanatory text has been added to the commentary in Fig. 5 (new 6). In addition, the discussion section has been revised and supplemented.
Thank you on behalf of all authors for the accurate and exact review of our manuscript.
Reviewer 3 Report
-There some typos:
(1) Line 125~126. Is it a title? If yes, it is suggested to be numbered.
(2) Line 118, what is “(2) (6)”? There are similar mistakes in other places.
(3) Line 104~108, the intent is not the same.
(4) The styles of symbols in the text should be the same as those in figures.
-It is suggested to be compared with the existing design methods to show the contribution of this study. It is not enough if only show the consistency between calculation and experiment simply. The method should be evaluated more comprehensively. For example, electromagnetic interference, etc. as discussed in line 57~61.
-For this study, is there any limitation or assumption?
Author Response
Reviewer 3
Comments to the Authors
-There some typos:
(1) Line 125~126. Is it a title? If yes, it is suggested to be numbered.
(2) Line 118, what is “(2) (6)”? There are similar mistakes in other places.
(3) Line 104~108, the intent is not the same.???
(4) The styles of symbols in the text should be the same as those in figures.
-It is suggested to be compared with the existing design methods to show the contribution of this study. It is not enough if only show the consistency between calculation and experiment simply. The method should be evaluated more comprehensively. For example, electromagnetic interference, etc. as discussed in line 57~61.
-For this study, is there any limitation or assumption?
To Reviewer 3:
Thank you for your review and valuable remarks.
-There some typos:
(1) Line 125~126. Is it a title? If yes, it is suggested to be numbered.
(2) Line 118, what is “(2) (6)”? There are similar mistakes in other places.
(3) Line 104~108, the intent is not the same.???
(4) The styles of symbols in the text should be the same as those in figures.
- The manuscript has been edited. The necessary editorial corrections have been made, according to the remarks.
-It is suggested to be compared with the existing design methods to show the contribution of this study. It is not enough if only show the consistency between calculation and experiment simply. The method should be evaluated more comprehensively. For example, electromagnetic interference, etc. as discussed in line 57~61.
- Thank you very much for the comment! As the power circuit does not consume current from the power supply during the whole half-period, this leads to poor indicators in terms of electromagnetic compatibility. In this aspect, in order for the system to be fully operational and to meet EMC standards, additional filters or active power factor correction (PFC) must be added. These and other clarifications have been added in the discussion section.
-For this study, is there any limitation or assumption?
- Thank you very much for the comment. The limitations and shortcomings of the scheme in general are commented in the editing of the manuscript in the discussion section. Generally speaking, they are related to the fact that: no energy is consumed during the entire half-period, and when operating in modes close to short circuit, the duration of consumption is further reduced; the circuit stabilizes the average value of the consumed current, but does not limit its maximum value; in addition, the issues already discussed to ensure electromagnetic compatibility standards; and other shortcomings in general of resonant inverters, which have already been commented in the introduction section.
Thank you on behalf of all authors for the accurate and exact review of our manuscript.
Reviewer 4 Report
The paper is interest and about a converter for charging ev applications.
It is importante to clarify the contibution. What are the main advantagens and the drawbacks of the proposal? Add references of the last 2 years.
It is mandatory to present simulations and experimental results of the dynamic behavior of the proposal. And, making a comparison with another solution presented in the literature.
The presented information does not allow to evaluate the viability of the proposal.
Author Response
Reviewer 4
Comments to the Authors
The paper is interest and about a converter for charging ev applications.
It is importante to clarify the contibution. What are the main advantagens and the drawbacks of the proposal? Add references of the last 2 years.
It is mandatory to present simulations and experimental results of the dynamic behavior of the proposal. And, making a comparison with another solution presented in the literature.
The presented information does not allow to evaluate the viability of the proposal.
To Reviewer 4:
Thank you for your review and valuable remarks.
It is importante to clarify the contibution. What are the main advantagens and the drawbacks of the proposal? Add references of the last 2 years.
- Thank you very much for the comment. The limitations and shortcomings of the scheme in general are commented in the editing of the manuscript in the discussion section. Generally speaking, they are related to the fact that: no energy is consumed during the entire half-period and when operating in modes close to short circuit, the duration of consumption is further reduced; the circuit stabilizes the average value of the consumed current, but does not limit its maximum value; in addition, the issues already discussed to ensure electromagnetic compatibility standards; and other shortcomings in general of resonant inverters, which have already been commented in the introduction section. New references has also been added.
It is mandatory to present simulations and experimental results of the dynamic behavior of the proposal. And, making a comparison with another solution presented in the literature.
The presented information does not allow to evaluate the viability of the proposal.
- When editing the manuscript, comparisons were made with other similar schemes. The study of dynamics is not the subject of this work and will be done in future work. On the other hand, given studies of energy dosing schemes for other applications with operation under significant load changes, such as induction heating, they should have very dynamic properties, taking into account their self-matching capabilities.
Thank you on behalf of all authors for the accurate and exact review of our manuscript.
Reviewer 5 Report
This article introduces a power converter for electric vehicles, which is an interesting research topic. There are some suggestions and comments for authors to revise their papers, as shown below.
(1) It is recommended that authors improve the abstract to show the advantages and main contributions of the method they proposed in this article, as well as some research results.
(2) Please further explain that the transistor operates under zero on and off current (ZCS), as shown in Figure 2.
(3) In line 137, for clarity, please further explain the parameter RE.
(4) In Equation (10), what's the value of w0/w in the design of the presented circuit?
(5) The title of Figure 5.-(1) is suggested to modify as "full-bridge current fed inverter".
(6)In the experimental setup shown in Figure 6, please include the circuit parameters of the "INVERTOR" in the additional table.
(7) In order to demonstrate that the switches of semiconductor devices in the circuit have the characteristics of ZCS and ZVS, it is recommended to include some experimental waveforms of power transistors, such as transistor voltage and transistor current, in the revised version.
Author Response
Reviewer 5
Comments to the Authors
This article introduces a power converter for electric vehicles, which is an interesting research topic. There are some suggestions and comments for authors to revise their papers, as shown below.
(1) It is recommended that authors improve the abstract to show the advantages and main contributions of the method they proposed in this article, as well as some research results.
(2) Please further explain that the transistor operates under zero on and off current (ZCS), as shown in Figure 2.
(3) In line 137, for clarity, please further explain the parameter RE.
(4) In Equation (10), what's the value of w0/w in the design of the presented circuit?
(5) The title of Figure 5.-(1) is suggested to modify as "full-bridge current fed inverter".
(6)In the experimental setup shown in Figure 6, please include the circuit parameters of the "INVERTOR" in the additional table.
(7) In order to demonstrate that the switches of semiconductor devices in the circuit have the characteristics of ZCS and ZVS, it is recommended to include some experimental waveforms of power transistors, such as transistor voltage and transistor current, in the revised version.
To Reviewer 5:
Thank you for your review and valuable remarks.
(1) It is recommended that authors improve the abstract to show the advantages and main contributions of the method they proposed in this article, as well as some research results.
- The abstract has been edited by adding the main advantages and disadvantages of the proposed scheme.
(2) Please further explain that the transistor operates under zero on and off current (ZCS), as shown in Figure 2.
- Thank you very much for the comment. Text has been added under Fig. 2, where this basic property is emphasized.
(3) In line 137, for clarity, please further explain the parameter RE.
- A text has been added explaining what is denoted by Re (Re is the equivalent resistance between points b and c of Fig. 1).
(4) In Equation (10), what's the value of w0/w in the design of the presented circuit?
- The value is added when correcting the manuscript.
(5) The title of Figure 5.-(1) is suggested to modify as "full-bridge current fed inverter".
- Thank you very much for the comment. It has been corrected in the text during the revision.
(6)In the experimental setup shown in Figure 6, please include the circuit parameters of the "INVERTOR" in the additional table.
- Thank you very much for the comment. The parameters were added to the description of the experimental bench.
(7) In order to demonstrate that the switches of semiconductor devices in the circuit have the characteristics of ZCS and ZVS, it is recommended to include some experimental waveforms of power transistors, such as transistor voltage and transistor current, in the revised version.
- Thank you very much for the suggestion. New figures (10a and 10b) have been added, showing oscillograms from experimental studies.
Thank you on behalf of all authors for the accurate and exact review of our manuscript.
Round 2
Reviewer 2 Report
The authors have carefully addressed the comments.
Reviewer 3 Report
The author has revised the manuscript according to the comments finely.
Author Response
First of all, we would like to thank you for the thorough review of our paper (energies-1465722) and the useful remarks to improve it.
Reviewer 3
Comments to the Authors
The author has revised the manuscript according to the comments finely.
To Reviewer 3:
Thank you very much for your review and valuable remarks. Your remarks and comments were very useful to us and will also help us in our future work on the development and deepening of our research.
Reviewer 4 Report
The authors try to change the paper in accordance with the comments of this reviewer but they do not answer the questions fully. As example: the dynamic behavior of the proposal in simulated and experimental results or the comparison with another methods using results.
Author Response
First of all, we would like to thank you for the thorough review of our paper (energies-1465722) and the useful remarks to improve it.
Reviewer 4
Comments to the Authors
The authors try to change the paper in accordance with the comments of this reviewer but they do not answer the questions fully. As example: the dynamic behavior of the proposal in simulated and experimental results or the comparison with another methods using results.
To Reviewer 4:
The proposed scheme is also successfully used for contactless dynamic charging of electric vehicles. The main advantage of energy dosing schemes is that the power does not depend on the size of the load. Therefore, during driving, when the equivalent load is constantly changing (due to the coefficient of magnetic coupling), the power transferred to the vehicle is constant. This text has been added to the conclusion section when editing the manuscript. On the other hand, the application of these dynamic charging schemes is not the subject of this study, as it is considered in our other works such as: Madzharov, N.; Hinov, N. Flexibility of Wireless Power Transfer Charging Station Using Dynamic Matching and Power Supply with Energy Dosing. Appl. Sci. 2019, 9, 4767. https://doi.org/10.3390/app9224767
On the other hand, the study of the sustainability of the whole system and the control synthesis is in the focus of our research interests and we hope to offer a manuscript dedicated to this topic soon.
Reviewer 5 Report
The authors have already revised their paper according to the reviewer's comments and suggestions including improvement of the abstract, explaination of the parameter RE, correction of the title of Figure 5.-(1)c, inclusion of the circuit parameters of the "INVERTOR" and some oscillograms from experimetnal studies. Therefore, it can be accepted in its current form. Congratulations to the authors!:)
Author Response
First of all, we would like to thank you for the thorough review of our paper (energies-1465722) and the useful remarks to improve it.
Reviewer 5
Comments to the Authors
The authors have already revised their paper according to the reviewer's comments and suggestions including improvement of the abstract, explaination of the parameter RE, correction of the title of Figure 5.-(1)c, inclusion of the circuit parameters of the "INVERTOR" and some oscillograms from experimetnal studies. Therefore, it can be accepted in its current form. Congratulations to the authors!:)
To Reviewer 5:
Thank you very much for your review and valuable remarks and for the appreciation of our efforts to improve the quality of the manuscript. Your remarks and comments were very useful to us and will also help us in our future work on the development and deepening of our research.
Round 3
Reviewer 4 Report
The changes are ok.